# Effect of Shearing and Heat Milling Treatment Temperature on the Crystallinity, Thermal Properties, and Molecular Structure of Rice Starch

**DOI:** 10.3390/foods12051041

**Published:** 2023-03-01

**Authors:** Naoto Suzuki, Marin Abiko, Hiroko Yano, Tomonori Koda, Akihiro Nishioka, Naoko Fujita

**Affiliations:** 1Laboratory of Plant Physiology, Department of Biological Production, Faculty of Bioresource Science, Akita Prefectural University, Akita City 010-0195, Akita, Japan; m23g011@akita-pu.ac.jp; 2Graduate School of Organic Materials Science, Yamagata University, Yonezawa City 992-8510, Yamagata, Japan; t211427d@st.yamagata-u.ac.jp (M.A.); hiroko.yano@yz.yamagata-u.ac.jp (H.Y.); koda@yz.yamagata-u.ac.jp (T.K.); nishioka@yz.yamagata-u.ac.jp (A.N.)

**Keywords:** rice starch, shear and heat, gel permeation chromatography, starch crystallinity, gelatinization, starch structure, amylopectin clusters, chain length distribution

## Abstract

Rice flour is produced by various methods for use in the food industry, but little is known about how the structure of starch is affected during rice flour production. In this study, the crystallinity, thermal properties, and structure of starch in rice flour were investigated after treatment with a shearing and heat milling machine (SHMM) at different temperatures (10–150 °C). Both the crystallinity and gelatinization enthalpy of starch showed an inverse relationship with the treatment temperature; rice flour treated with the SHMM at higher temperatures showed lower crystallinity and gelatinization enthalpy than that treated at lower temperatures. Next, the structure of undegraded starch in the SHMM-treated rice flour was analyzed by gel permeation chromatography. A significant reduction in the molecular weight of amylopectin was observed at high treatment temperatures. Chain length distribution analysis showed that the proportion of long chains (degree of polymerization (DP) > 30) in rice flour decreased at temperatures ≥ 30 °C. By contrast, the molecular weight of amylose did not decrease. In summary, the SHMM treatment of rice flour at high temperatures resulted in starch gelatinization, and the amylopectin molecular weight decreased independently, due to the cleavage of amorphous regions connecting the amylopectin clusters.

## 1. Introduction

Wheat, the third most highly produced cereal grain in the world after maize and rice, is used worldwide as flour to prepare a variety of foods, such as bread, noodles, and confectioneries, and plays a paramount role in human nutrition [1]. However, a certain percentage of people is allergic to gluten, and non-gluten food sources such as rice are gaining popularity as alternatives to wheat [2]. Rice, which is also consumed as a grain in Asian countries, has long been used as flour for making noodles and other food products, and more recently for making breads and confectioneries [1]. Rice flour is produced using a wide variety of milling methods that involve the use of heat, warm water, a ball mill, high pressure, microwaves, extruders, or γ-radiation [3,4,5,6,7,8,9,10], and the degree of crystallinity of the resulting rice flour varies with the milling method. A shearing and heat milling machine (SHMM) has been proposed as an easy method for modifying rice crystallinity [11]. The SHMM applies mechanical shearing force and heat during the milling process, producing amorphous rice flour at temperatures > 80 °C without requiring the addition of water. SHMMs are characterized by their ability to continuously produce amorphous rice flour in a short time [11]. In general, bread prepared solely from rice flour is less volumetric than wheat bread; however, mixing amorphous rice flour produced using the SHMM method would increase the proportion of air spaces in the dough, and consequently the volume of rice-flour bread [12].

Previously, an SHMM was used to investigate how different factors including shearing temperature, the gap between millstones, and grain moisture content, affect the crystallinity, physicochemical properties, and structure of starch [13]. When polished rice grains with 12% moisture content were treated in the SHMM at 120 °C, with a 10 µm gap between millstones, the crystallinity of the starch was lost completely. By contrast, when the treatment temperature and rice grain moisture content were reduced to 15 °C and 1%, respectively, and the gap between the millstones was increased to 100 µm, the crystallinity of starch was retained to a certain level. Thus, the SHMM can control the degree of crystallinity by controlling the temperature of mortar, speed of rotations, and distance between millstones.

When the starch of completely amorphous rice flour was subjected to gel permeation chromatography (GPC) using a Sephacryl S-1000 column without any enzymatic degradation, the amylopectin peak was lower than that of crystalline rice flour, whereas the amylose peak was larger, indicating that high-molecular-weight amylopectin was converted into low-molecular-weight amylopectin [13]. GPC performed using the Sephacryl S1000 column could separate only two peaks, corresponding to amylopectin and amylose, and the low-molecular-weight amylopectin peak overlapped with the amylose peak. Therefore, it was difficult to clearly identify the structural changes in amylopectin and amylose.

In previous studies, when undegraded starch was separated by GPC and detected with a refractive index (RI) detector, it could be separated into only two peaks, one each for amylopectin and amylose [14,15,16,17]; the separation of undegraded starch into three or more peaks has not been reported to date. Recently, we developed a gel filtration method that could separate unresolved starch into high-molecular-weight amylopectin, low-molecular-weight amylopectin, and amylose [18]. Additionally, when debranched starch is subjected to another series of GPC, it is possible to clearly determine the ratio of amylose content to amylopectin content [19]. Furthermore, debranched starch can be separated by capillary electrophoresis according to the DP, making it possible to determine the fine structure of amylopectin [20].

In this study, we investigated the crystallinity and gelatinization enthalpy of rice flour treated with an SHMM at various temperatures. Simultaneously, to determine the effect caused by SHMM treatment temperature on starch structure, the following three experiments were performed in this study: (1) the degree of reduction in the molecular weight of amylopectin was examined by the GPC of undegraded starch using a series of columns (Toyopearl HW75S × 2-HW65S-HW55S), (2) the ratio of amylose and amylopectin was examined by GPC of debranched starch using another series of columns (Toyopearl HW55S-HW50S × 3), and (3) further structural analysis of debranched starch was performed by capillary electrophoresis to identify the cleavage sites in starch molecules. The relationships among the crystallinity, gelatinization enthalpy, and structure of starch in rice flour sheared at different temperatures are discussed.

## 2. Materials and Methods

### 2.1. Rice and Milling Conditions

A non-glutinous japonica rice cultivar, ‘Haenuki’, harvested in Yamagata (Japan) in 2018 was used in this study. ‘Haenuki’ is a staple rice cultivar developed in Yamagata Prefecture and is widely consumed as cooked rice, because of its good taste, at both domestic and commercial scales. Before milling, a commercial rice polishing machine (VP-32; Yamamoto Co., Ltd., Tendo, Japan) was used to polish ‘Haenuki’ grains until their weight was ca. 90% of that of unpolished rice grains. Six rice flour samples (A–E) were examined in this study (Table 1). Sample A (crystalline rice flour) was produced using an air flow pulverizer (MP2-350YS2, Yamamoto Co., Ltd., Tendo, Japan). Air flow milling does not damage the majority of starch granules. Samples B–F were obtained using a commercial millstone-type SHMM (KGW-G015; 250 mm wide, 200 mm deep, and 370 mm high; West Co., Ltd., Nagaoka, Japan) [11]. A thermostat ring heater in the upper millstone allows grinding by rotating the lower mortar at a controlled temperature. The temperature of the upper millstone was controlled with a ring heater when milling at high temperatures (100 °C, 120 °C, and 150 °C), and with a chiller when milling at low temperatures (10 °C and 30 °C). The gap between millstones and the moisture content of rice grains were maintained at 10 µm and 13.9%, respectively.

### 2.2. Wide-Angle X-ray Diffraction

The crystal diffraction pattern of the rice flour samples was examined by measuring wide-angle X-ray diffraction (WAXD) using an X-ray diffractometer (Ultima IV; Rigaku Co., Ltd., Tokyo, Japan), with a Cu-Kα radiation source at a wavelength of 0.154 nm. The tube voltage and current were maintained at 40 kV and 40 mA, respectively. Samples were scanned at room temperature from 5° to 35° at a scan rate of 10°/min. The crystallinity of the samples was measured using the peak separation method.

The peaks obtained from the WAXD measurements were separated into crystalline reflection and amorphous scattering using the Peak Fit software (ver. 4.12; Sea Solve Software, Inc., Framingham, USA). The peaks from crystalline reflection were defined as diffraction peaks that appeared at 15°, 17°, 18°, and 23°. The area of amorphous scattering was specified as a broad peak that covered the diffraction image. The crystallinity of a sample (Cx, %) was calculated using the following equation:Cx = Sc/(Sc + Sa) × 100,(1)
where Sc represents the integrated value of the peak area due to crystal reflection, and Sa represents the integrated value of the peak due to amorphous scattering.

### 2.3. Analysis of the Thermal Properties of Starch

The thermal properties of rice starch were measured by differential scanning calorimetry (DSC) (DSC 8500; Perkin Elmer Inc., Waltham, MA, USA). Briefly, 10 mg of rice flour and 30 μL of deionized water were sealed in a stainless steel pan. The temperature was increased from 25 °C to 95 °C at a rate of 5 °C/min. Empty pans were used as a reference. The resulting endothermic peaks were analyzed using the Pyris software (Perkin Elmer Inc., Waltham, USA), and the gelatinization transition temperatures (T_o_, onset temperature; T_p_, peak temperature; T_c_, conclusion temperature) and gelatinization enthalpy (ΔH) were determined.

### 2.4. GPC of Undegraded Starch

Rice flour samples (20 mg) were suspended in 1.6 mL of distilled water in a 15 mL centrifuge tube and gelatinized with 400 µL of 5 M NaOH for 30 min at 37 °C. Then, 2 mL of distilled water and 2 mL of eluent (0.2% NaCl/0.05 M NaOH) were added to each sample, followed by filtration through a 5 µm Durapore polyvinylidene fluoride (PVDF) membrane (Merck, Darmstadt, Germany). A 5 mL aliquot of the filtrate was used in a 4-column system (Toyopearl HW75S × 2-HW65S-HW55S; 2.2 cm diameter, 30 cm × 4; Tosoh Corp., Tokyo, Japan) that was pre-equilibrated with the eluent, and carbohydrates were detected using an RI detector (RI-8020; Tosoh Corp.). The columns were incubated at 40 °C, and the samples were eluted at a flow rate of 1 mL/min using a DP-8020 pump (Tosoh Corp.).

### 2.5. GPC of Debranched Starch

GPC analysis of the debranched starch extracted from the rice flour samples was performed using Toyopearl HW55S-HW50S × 3 columns (2.2 cm diameter, 30 cm × 4; Tosoh Corp.), as described by Toyosawa et al. [19].

### 2.6. Chain Length Distribution Analysis of Rice Starch

The chain length distribution of starch in the rice flour samples was estimated by fluorophore-assisted capillary electrophoresis (FACE) with a P/ACE MDQ Carbohydrate System (Sciex, Framingham, MA, USA), according to the methods of O’Shea and Morell [21] and Fujita et al. [20] The difference in chain length distribution was expressed as the molar change (%).

## 3. Results

### 3.1. Effect of SHMM Treatment Temperature on the Crystallinity of Starch

Figure 1 shows the WAXD results. The crystallinity of the ‘Haenuki’ flour samples subjected to air flow milling and SHMM treatment at different temperatures (10–150 °C) was analyzed using the WAXD measurements, and the degree of crystallinity was calculated. The samples subjected to air flow milling showed typical WAXD patterns of type A starch, with crystallinity peaks at 2θ = 15°, 17°, 18°, and 23°. These four peaks were also observed in the rice flour samples treated with the SHMM at 10 °C and 30 °C, but the peak heights were lower than air flow milling in the following order: air flow milling rice flour > 10 °C > 30 °C (Figure 1). By contrast, the rice flour samples exposed to higher temperatures showed significantly lower crystallinity (4.3% at 100 °C, 1.7% at 120 °C, and 1.2% at 150 °C). The rice flour samples treated with the SHMM at 120 °C and 150 °C almost completely lost their peaks, similar to those treated with SHMM at 120 °C at 12% moisture content and a 10 μm gap between millstones [13]. These results suggest that crystallinity is maintained to some extent when the temperature during the SHMM treatment is 30 °C or lower, but it decreases rapidly at temperatures ≥ 100 °C.

### 3.2. Effect of SHMM Treatment Temperature on the Thermal Properties of Starch

Next, the thermal properties of each rice flour sample were measured by DSC (Figure 2). The T_o_, T_p_, and T_c_ of rice flour subjected to air flow milling were 62.5 °C, 69.0 °C, and 74.7 °C, respectively, and the gelatinizing enthalpy (ΔH) was 7.7 J/g (Figure 2). The T_o_, T_p_, and T_c_ of rice flour samples treated with the SHMM at 10 °C (62.0 °C, 67.4 °C, and 71.4 °C, respectively) were slightly lower than those of the samples treated with air flow milling, consistent with the trend observed previously [13]. This may be because the double helix in amylopectin clusters was partially gelatinized and unwound by the SHMM treatment, and the remaining double helix in the SHMM-treated samples was slightly shorter than that of the air flow-milled samples. Compared with rice flour subjected to air flow milling, the ΔH of rice flour treated with the SHMM was considerably lower at 10 °C (2.5 J/g) and even lower at 30 °C (2.1 J/g). The T_o_, T_p_, and T_c_ of rice flour samples treated with the SHMM at 30 °C were similar to those treated with the SHMM at 10 °C (data not shown). The ΔH of rice flour treated with the SHMM above 100 °C was not detected (Figure 2).

### 3.3. SHMM Treatment Temperature Effect on Starch Molecular Weight

To investigate the effects of different temperatures (10–150 °C) on the structure of starch in rice flour treated with SHMM, the undegraded starch in the SHMM-treated samples was separated by GPC using Toyopearl HW75S × 2-HW65S-HW55S columns and compared with starch in the rice flour subjected to air flow milling (Figure 3).

Previously, undegraded starch extracted from the japonica rice cultivar ‘Nipponbare’ was separated into three peaks (Peaks 1, 2, and 3) using Toyopearl HW75S × 2-HW65S-HW55S columns [18]; Peaks 1 and 2 comprised mainly high- and low-molecular-weight amylopectin, respectively, while Peak 3 comprised amylose and even lower molecular weight amylopectin [18]. In the current study, when rice flour prepared by air flow milling was applied to the same columns, three peaks were detected, as in the case of purified ‘Nipponbare’ starch. The percentage of each peak obtained by the GPC of undegraded starch is shown in Appendix A. Peak 1 contained the highest percentage (43.2%) among the three peaks, followed by Peak 2 (41.2%) (Figure 3; Appendix A). The GPC pattern of rice flour treated with the SHMM at 10 °C had a slightly higher percentage of Peak 1 (46.2%) than that of rice flour subjected to air flow milling, although the difference was not significant (Appendix A). Rice flour treated with the SHMM at 30 °C showed a significant reduction in Peak 1 (32.2%) and a significant increase in Peak 2 (50.9%); rice flour treated with the SHMM at 100 °C showed a drastic decrease in Peak 1 (6.3%) and a significant increase in Peak 2 (65.7%) and Peak 3 (28.0%); rice flour treated with the SHMM at 120 °C and 150 °C showed almost no Peak 1, a decrease in Peak 2 (52.3 and 56.0%, respectively), and a significant increase in Peak 3 (46.6% and 43.8%, respectively) compared with rice flour treated with the SHMM at 100 °C (Figure 3; Appendix A). We speculate that the amylopectin detected in Peak 2 decreased in molecular weight and was detected in Peak 3.

### 3.4. Effect of SHMM Treatment Temperature on the Structure of Debranched Starch

Next, to compare the structure of debranched starch in rice flour treated with air flow milling and the SHMM at 10–150 °C, GPC was performed using Toyopearl HW55S-HW50S × 3 columns (Appendix A; Table 2). The GPC pattern of debranched starch in air flow-milled rice flour showed the following three peaks: Frac. I (amylose), Frac. II (long amylopectin chains), and Frac. III (short amylopectin chains) [19]. The percentage of Frac. I was 18.4%, and the ratio of short-chain to long-chain amylopectin (Frac. III/Frac. II) was 2.4 (Table 2). The Frac. I percentage and the Frac. III/Frac. II ratio of rice flour treated with the SHMM at 10–150 °C ranged from 16.2 to 19.5% and 2.1 to 2.6, respectively, and multiple comparisons using the Tukey–Kramer method showed no significant differences among any of the rice flour samples, including that subjected to air flow milling (Table 2).

### 3.5. Effect of SHMM Treatment Temperature on the Chain Length Distribution of Starch

To analyze the differences in the fine structure of amylopectin in each rice flour sample, the chain length distribution of starch was analyzed by FACE (Figure 3A). The chain length distribution patterns of all the rice flour samples were almost identical at first glance (Figure 4A). To examine the GPC patterns in further detail, the difference in percentage molar change (ΔMolar [%]/Molar [%] × 100) between the rice flour samples treated with the SHMM at 10–150 °C and those subjected to air flow milling was calculated and plotted (Figure 4B). Rice flour treated with the SHMM at any temperature showed a decrease at DP > 30; the rice flour treated with the SHMM at 10 °C showed a decrease of approximately 5% at DP 70; the rice flour treated with the SHMM at temperatures higher than 30 °C showed a decrease of 10–20% (Figure 4B).

## 4. Discussion

Starch crystallinity is governed by the molecular structure of amylopectin. For example, the crystallinity and ΔH of α-glucans in *sugary-1* (isoamylase 1 deficient mutant) rice lines, which partially accumulate phytoglycogen with highly branched and short side chains, are lower than those of wild-type rice, which accumulates starch [16,22,23]. This is because the side chains of phytoglycogen are too short to form a double helix, which is responsible for crystallinity and ΔH. In this study, we examined the changes in the crystallinity and ΔH of starch caused by the physical treatment of rice flour with an SHMM at 10 °C, 30 °C, 100 °C, 120 °C, and 150 °C. The reason why these temperature conditions were chosen is because setting a stable temperature for the mortar in SHMM is difficult if the temperature is higher than room temperature (~30 °C) and lower than 100 °C. The results showed that SHMM treatment decreased the crystallinity of starch to a greater extent than the air flow milling method, and that the degree of reduction in crystallinity was greater at higher temperatures than at lower temperatures, resulting in almost zero crystallinity at temperatures above 100 °C (Figure 1). The ΔH of rice flour treated with the SHMM, as analyzed by DSC, was also lower than that of rice flour subjected to air flow milling, and no ΔH was detected in rice flour treated with the SHMM at temperatures above 100 °C (Figure 2). Additionally, a strong correlation was observed between the crystallinity (measured by WAXD) and ΔH (measured by DSC) of the rice flour samples treated with the SHMM at various temperatures. The crystallinity of starch is derived mainly from amylopectin [24], and adjacent amylopectin side chains form double helices. By contrast, the ΔH of starch indicates an endothermic reaction that occurs when the double helix is unwound by gelatinization [25]. Therefore, the reduction in the crystallinity and ΔH of starch during the SHMM treatment could be attributed to its gelatinization. The reason why the crystallinity and ΔH of the samples treated with SHMM decreased, even when the temperature was set at 10 °C or 30 °C below the gelatinization temperature, compared with those of the samples subjected to airflow milling is thought to be because the temperature of the mortar, which was in direct contact with the rice flour, was higher than the set temperature, causing the gelatinization of a small proportion of the starch. The rice flour treated with the SHMM at 120 °C with 12% moisture content and a 10 μm gap between millstones was completely amorphous, whereas the rice flour treated with the SHMM under a reduced moisture content of 1% remained crystalline, probably because of the lack of moisture required for gelatinization [13]. The polished rice grains used in this study had a moisture content of 13.9%, which is sufficient for the gelatinization of starch.

Next, we attempted to separate the undegraded starch in the rice flour samples by GPC. Rice flour treated with the SHMM at a higher temperature showed a significant reduction in the molecular weight of amylopectin (Figure 3; Appendix A). Previously, the peak-top of molecular weights of purified ‘Nipponbare’ starch in Peaks 1, 2, and 3 examined using the same GPC columns (Toyopearl HW75S-HW65S-HW55S × 2) as in this study were estimated to be 6.0 × 10^8^, 3.4 × 10^7^, and 8.4 × 10^5^, respectively, with DP values of 19,480, 1120, and 27, respectively [18]. Therefore, amylopectin with a DP of ~20,000 (Peak 1) changed to ~1/20 (Peak 2) after SHMM treatment. In the rice flour samples treated with SHMM at temperatures > 100 °C, the DP of ~1000 (Peak 2) decreased to ~1/40 (Peak 3). By contrast, when the starch in each rice flour sample was debranched and analyzed by GPC using the Toyopearl HW55S-HW50S × 3 columns, the SHMM treatment temperature had no effect on the apparent amylose content (Table 2) and chromatography patterns (Appendix A). This indicates that the molecular weight of linear amylose is not affected by SHMM treatment even at high temperatures.

Murakami et al. [13] separated the undegraded starch of crystalline rice flour and SHMM-treated rice flour into two peaks (amylopectin and amylose) using GPC columns that contained Sephacryl S-1000. The peak that corresponded to the amylopectin fraction was examined its detail by FACE after debranching. The molar change (%) in the amylopectin chain length above DP35 in the SHMM-treated rice flour was lower than that in crystalline rice flour [13]. In this study, the ratio of short-chain to long-chain amylopectin, as determined by the GPC of debranched starch, showed no significant difference between the rice flour samples subjected to air flow milling and those treated with the SHMM (Table 2), but detailed analysis of the chain length distribution revealed that the rice flour samples treated with the SHMM at temperatures higher than 30 °C had a lower proportion of long chains (DP ≥ 30) connecting amylopectin clusters. This may be because of the SHMM-induced physical cleavage of the long B_2–3_ chains connecting amylopectin clusters.

These results strongly suggest that the shearing-induced cleavage of amylopectin molecules does not occur within the amylopectin clusters, but rather in the amorphous regions connecting the clusters. Therefore, the SHMM treatment results in the reduction in the molecular weight of amylopectin molecules. It was also found that the shearing-induced reduction in the molecular weight of amylopectin was more severe at higher processing temperatures. The gelatinization of starch, which reduces crystallinity and ΔH, as well as the reduction in the molecular weight of amylopectin, are thought to occur simultaneously but independently, and the degree of both of these processes is greater at higher temperatures than at lower temperatures.

A vast amount of research has been conducted on the processing of gluten-free bread using flour prepared from rice and other grains, except wheat [1]. Generally, gluten-free bread made from crystalline rice flour alone has less volume and an inferior taste. To solve these problems, amorphous rice flour, which is obtained through physical shearing by SHMM or other methods, is added [12]. In terms of bread volume, bread in which 15% of wheat flour was substituted with rice puree was comparable with that containing 100% wheat flour, although the volume of the bread that contained rice flour was small. Interestingly, the ground rice puree obtained by shearing was less viscous and easier to handle during processing than the conventional rice puree, and the bread volume was greater [26]. Ground rice puree obtained by shearing contained low-molecular-weight amylopectin, which may explain its reduced viscosity [26]. In future studies, the physical properties and molecular structure of starch must be considered as important indicators of processing suitability and taste when producing gluten-free rice-based foods.

## 5. Conclusions

In this study, we found that rice flour samples treated with an SHMM at different temperatures showed a lower degree of crystallinity and lower enthalpy, because of gelatinization, compared with the samples treated with air flow milling. The extent of the reduction in the degree of crystallinity and enthalpy of SHMM-treated samples was more pronounced at higher temperatures. Furthermore, the degree of degradation of amylopectin and amylose, which could not be clarified previously [13], was determined in this study using a combination of two GPC systems. The SHMM treatment caused almost no reduction in the molecular weight of amylose, but substantially reduced the molecular weight of amylopectin, which has a branched structure. The molecular weight of amylopectin in the samples treated with the SHMM at high temperatures was 1/20th to 1/800th of that of amylopectin in the untreated rice flour samples. The shearing-induced cleavage of amylopectin molecules occurred in amorphous regions, which connect amylopectin clusters.

## Figures and Tables

**Figure 1 foods-12-01041-f001:**
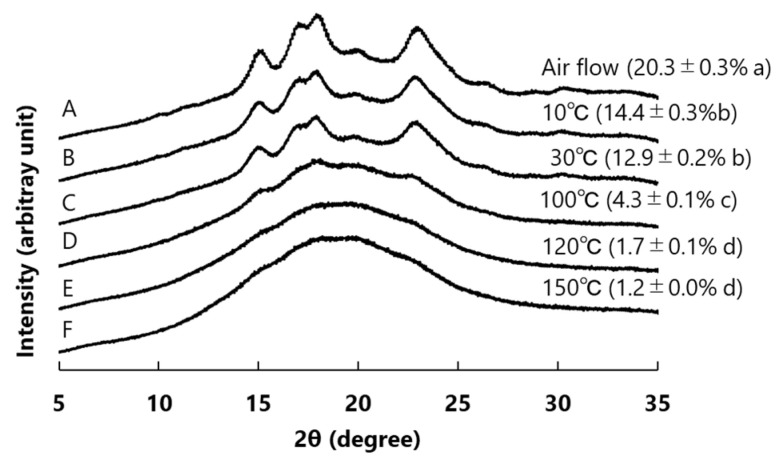
Wide-angle X-ray diffraction (WAXD) patterns and degree of crystallinity (%) of rice flour samples prepared by the SHMM treatment at different temperatures or by air flow milling. (**A**) Rice flour samples prepared by air flow milling. (**B**–**F**) Rice flour samples subjected to SHMM at 10 °C (**B**), 30 °C (**C**), 100 °C (**D**), 120 °C (**E**), and 150 °C (**F**). Percentages indicate the degree of crystallinity. Data represent mean ± standard error (SE; *n* = 3). Different lowercase letters (a–d) in the figure indicate significant differences, as determined by the Tukey–Kramer method (*p* ˂ 0.05).

**Figure 2 foods-12-01041-f002:**
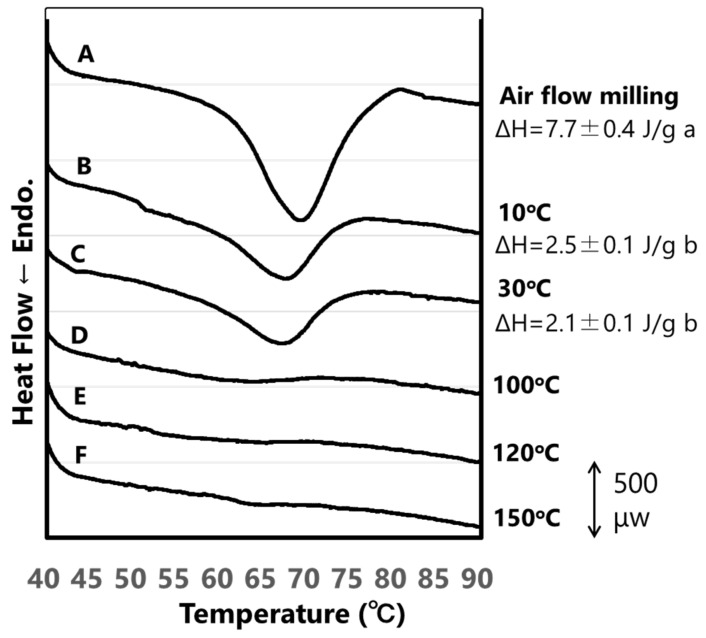
Effects of SHMM treatment temperature on the thermal properties of rice flour samples. (**A**) Rice flour prepared by air flow milling. (**B**–**F**) Rice flour samples subjected to SHMM at 10 °C (**B**), 30 °C (**C**), 100 °C (**D**), 120 °C (**E**), and 150 °C (**F**). Numbers indicate the gelatinization enthalpy (ΔH) of starch. Data represent mean ± SE (*n* = 3). Different lowercase alphabet letters (a–b) in the figure indicate significant differences, as determined by the Tukey–Kramer method (*p* ˂ 0.05). The ΔH of flour samples subjected to SHMM at 100 °C, 120 °C, and 150 °C could not be measured.

**Figure 3 foods-12-01041-f003:**
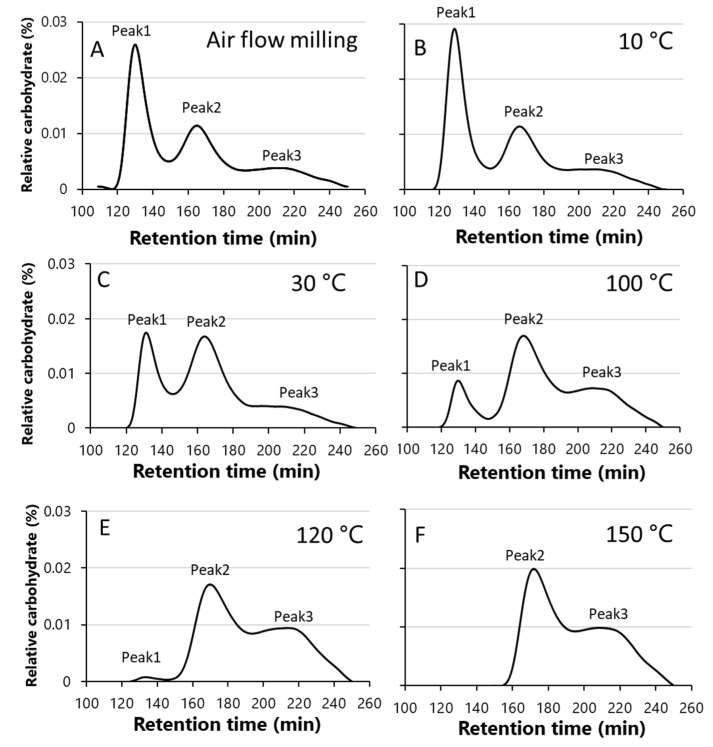
Typical GPC patterns (Toyopearl HW75S×2-HW65S-HW55S) of rice starch prepared by the SHMM treatment at different temperatures or by air flow milling. (**A**) Rice flour prepared by air flow milling. (**B**–**F**) Rice flour samples subjected to SHMM at 10 °C (**B**), 30 °C (**C**), 100 °C (**D**), 120 °C (**E**), and 150 °C (**F**). Peaks 1, 2, and 3 represent high-molecular-weight amylopectin, low-molecular-weight amylopectin, and amylose and lower molecular-weight amylopectin, respectively.

**Figure 4 foods-12-01041-f004:**
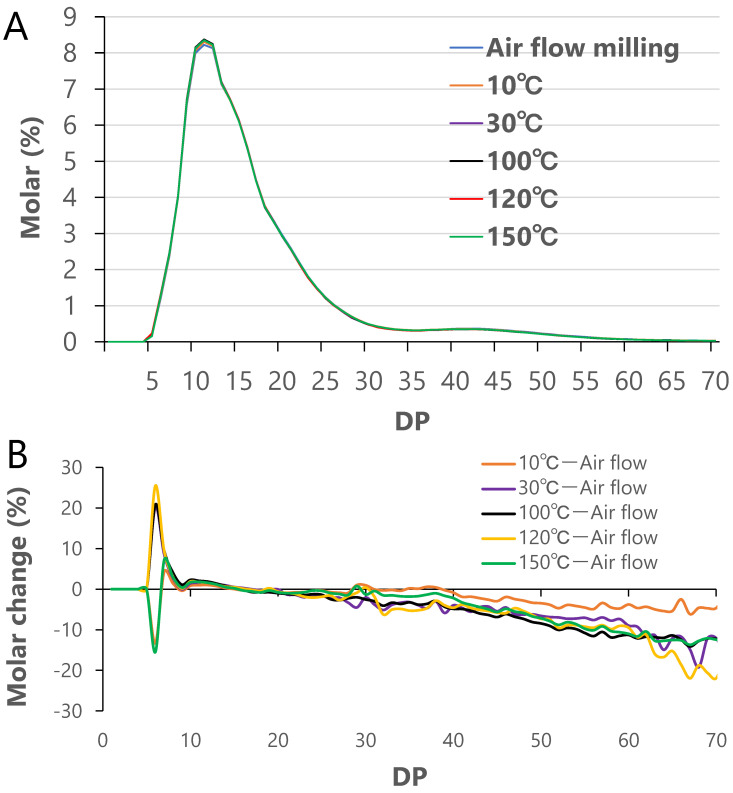
Chain-length distributions for rice flour prepared by SHMM treatment at different temperatures and crystalline rice flour by air flow milling as determined using the FACE method. (**A**) Molar percentage for each liberated chain to the total chains after debranchin starch. (**B**) Rate of molar change in chain length distribution for each rice flour by SHMM treatment at different temperature compared to air flow milling (ΔMolar [%]/Molar [%] × 100).

**Table 1 foods-12-01041-t001:** Rice Flour Samples Used in This Study.

Sample	Milling Type ^1^	Temperature
A	Air flow milling	-
B	SHMM	10 °C
C	SHMM	30 °C
D	SHMM	100 °C
E	SHMM	120 °C
F	SHMM	150 °C

^1^ SHMM: Shearing and milling machine.

**Table 2 foods-12-01041-t002:** Apparent Amylose Content and Ratio of Short-chain to Long-chain Molecules of Amylopectin in Various Rice Flour Samples.

Milling Method	Frac. I (%) ^1^	Frac. III/Frac. II ^1^
Air flow milling	18.4 + 1.9 a	2.4 + 0.1 a
SHMM at 10 °C	16.9 + 0.5 a	2.3 + 0.1 a
SHMM at 30 °C	16.2 + 0.5 a	2.3 + 0.1 a
SHMM at 100 °C	19.5 + 0.9 a	2.6 + 0.2 a
SHMM at 120 °C	17.0 + 1.9 a	2.4 + 0.0 a
SHMM at 150 °C	18.6 + 0.6 a	2.1 + 0.1 a

^1^ Frac. I indicates the apparent amylose content; Frac. III/Frac. II represents the ratio of short-chain to long-chain molecules of amylopectin (Appendix A). Data represent the mean ± SE of three replicates. Statistical analysis was carried out using the Tukey–Kramer method, and no significant difference was detected (*p* ˂ 0.05).

## Data Availability

Not applicable.

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
