# Peer review of "Effect of Shearing and Heat Milling Treatment Temperature on the Crystallinity, Thermal Properties, and Molecular Structure of Rice Starch"

_foods, 2023, doi:10.3390/foods12051041_

Round 1

Reviewer 1 Report

Rice starch is missing in the keywords section - it wasn't any starch that was the subject of the research, but rice starch.

A fairly well-written introduction, introducing the reader to the research material. however, would suggest correcting two things:

- a detailed description of the column lines 45-46 is completely unnecessary

- only in the part concerning the material I found out what kind of rice was used. From what I know as a European sushi gourmet, this variety is popularly used to make sushi. I think that it would be valuable in the introduction to briefly characterize - compare the quality of rice with others.

Perfectly matched analytical methods to understand changes in starch during different milling conditions.

Clear and comprehensive presentation of results. Well written discussion, although I have the impression that for obvious reasons (references to the results) some of the content is repeated. I know that the journal's requirement is to separate the results from the discussion, but more and more papers are accepted with a shared chapter Discussion and discussion of the results - I leave any changes to the authors' decision. Please write your conclusions

Author Response

I attached the file.

Reviewer 2 Report

This manuscript investigated the effects of a shearing and heat milling treatment at different temperature on the crystallinity, physicochemical properties, and structure of starch in rice. The manuscript has some important points, and the results are readable. It may be of potential interest to the reader. However, there are some flaws in the manuscript that affect the significance of the reported results. At present, modification of gluten-free starch raised worldwide attraction, while the novelty of SHMM is not well organized in this article. And the new finding of the experiment design seems to have used some new different columns to separate different molecular-weight amylopectin and amylose, if added more comparing groups, it will be more convincing and richer in content.

1.     Please explain the difference between the manuscript and the article the author published on 2017, ie: Effects of Shear and Heat Milling Treatment on Thermal Properties and Molecular Structures of Rice Starch, https://doi.org/10.1002/star.201700164”?

2.     The title “Effect of the Temperature of Shearing and Heat Milling Treatment on the Physicochemical Properties and Molecular Structure of Rice Starch”, about the physicochemical properties, in the article, only has the crystallinity and thermal properties of starch, is there more?

3.     From the introduction, it seems how different temperatures of SHMM affect the crystallinity, physicochemical properties, and structure of starch has been investigated, so the novelty of SHMM treatment and the use of new columns of GPC has not well clarified.

4.     The last paragraph of the discussion, doesn’t seem to have close relationship with the results, please add more link with the results or have a conclusion.

5.     What is the moisture of the rice starch? Does it affect the result in this manuscript?

6.     Why the temperature is set at 10 ℃, 30 ℃, 100℃,120℃ and 150℃?

7.     L298, “therefore” should change to “Therefore,”.

Author Response

I attached the file.

Round 2

Reviewer 1 Report

The authors referred to the comments in the previous review. The corrections and amendments made significantly improved the quality of work. The work may be published.